# DISTRIBUTED DISTRIBUTIONAL DETERMINISTIC POLICY GRADIENTS

**Gabriel Barth-Maron,** * **Matthew W. Hoffman,** * **David Budden, Will Dabney,**
**Dan Horgan, Dhruva TB, Alistair Muldal, Nicolas Heess, Timothy Lillicrap**
DeepMind
London, UK
{gabrielbm, mwhoffman, budden, wdabney, horgan, dhruvat,
 alimuldal, heess, countzero}@google.com

## ABSTRACT

This work adopts the very successful distributional perspective on reinforcement learning and adapts it to the continuous control setting. We combine this within a distributed framework for off-policy learning in order to develop what we call the Distributed Distributional Deep Deterministic Policy Gradient algorithm, D4PG. We also combine this technique with a number of additional, simple improvements such as the use of $N$-step returns and prioritized experience replay. Experimentally we examine the contribution of each of these individual components, and show how they interact, as well as their combined contributions. Our results show that across a wide variety of simple control tasks, difficult manipulation tasks, and a set of hard obstacle-based locomotion tasks the D4PG algorithm achieves state of the art performance.

## 1 INTRODUCTION

The ability to solve complex control tasks with high-dimensional input and action spaces is a key milestone in developing real-world artificial intelligence. The use of reinforcement learning to solve these types of tasks has exploded following the work of the Deep Q Network (DQN) algorithm (Mnih et al., 2015), capable of human-level performance on many Atari games. Similarly, ground breaking achievements have been made in classical games such as Go (Silver et al., 2016). However, these algorithms are restricted to problems with a finite number of discrete actions.

In control tasks, commonly seen in the robotics domain, continuous action spaces are the norm. For algorithms such as DQN the policy is only implicitly defined in terms of its value function, with actions selected by maximizing this function. In the continuous control domain this would require either a costly optimization step or discretization of the action space. While discretization is perhaps the most straightforward solution, this can prove a particularly poor approximation in high-dimensional settings or those that require finer grained control. Instead, a more principled approach is to parameterize the policy explicitly and directly optimize the long term value of following this policy.

In this work we consider a number of modifications to the Deep Deterministic Policy Gradient (DDPG) algorithm (Lillicrap et al., 2015). This algorithm has several properties that make it ideal for the enhancements we consider, which is at its core an off-policy actor-critic method. In particular, the policy gradient used to update the actor network depends only on a learned critic. This means that any improvements to the critic learning procedure will directly improve the quality of the actor updates. In this work we utilize a distributional (Bellemare et al., 2017) version of the critic update which provides a better, more stable learning signal. Such distributions model the randomness due to intrinsic factors, among these is the inherent uncertainty imposed by function approximation in a continuous environment. We will see that using this distributional update directly results in better gradients and hence improves the performance of the learning algorithm.

Due to the fact that DDPG is capable of learning off-policy it is also possible to modify the way in which experience is gathered. In this work we utilize this fact to run many actors in parallel, all feeding into a single replay table. This allows us to seamlessly distribute the task of gathering

---

*Authors contributed equally.

experience, which we implement using the ApeX framework (Horgan et al., 2018). This results in significant savings in terms of wall-clock time for difficult control tasks. We will also introduce a number of small improvements to the DDPG algorithm, and in our experiments will show the individual contributions of each component. Finally, this algorithm, which we call the Distributed Distributional DDPG algorithm (D4PG), obtains state-of-the-art performance across a wide variety of control tasks, including hard manipulation and locomotion tasks.

## 1.1 RELATED WORK

Historically, estimation of the policy gradient has relied on the likelihood ratio trick (see e.g. Glynn, 1990), more commonly known as REINFORCE (Williams, 1992) in the reinforcement learning community. Modern variants of these so-called "vanilla" policy gradient methods include the work of (Mnih et al., 2016). Alternatively, one can consider second-order or "natural" variants of this objective, a set of techniques that include e.g. the Natural Actor-Critic (Peters & Schaal, 2008) and Trust Region Policy Optimization (TRPO) (Schulman et al., 2015) algorithms. More recently Proximal Policy Optimization (PPO) (Schulman et al., 2017), which can be seen as an approximation of TRPO, has proven very effective in large-scale distributed settings. Often, however, algorithms of this form are restricted to learning on-policy, which can limit both the amount of data-reuse as well as restrict the types of policies that are used for exploration.

The Deterministic Policy Gradient (DPG) algorithm (Silver et al., 2014) upon which this work is based starts from a different set of ideas, namely the policy gradient theorem of (Sutton et al., 2000). The *deterministic* policy gradient theorem builds upon this earlier approach, but replaces the stochastic policy with one that includes no randomness. This approach is particularly important because it had previously been believed that the deterministic policy gradient did not exist in a model-free setting. The form of this gradient is also interesting in that it does not require one to integrate over the action space, and hence may require less samples to learn. DPG was later built upon by Lillicrap et al. (2015) who extended this algorithm and made use of a deep neural network as the function approximator, primarily as a mechanism for extending these results to work with vision-based inputs. Further, this entire endeavor lends itself very readily to an off-policy actor-critic architecture such that the actor's gradients depend only on derivatives through the learned critic. This means that by improving estimation of the critic one is directly able to improve the actor gradients. Most interestingly, there have also been recent attempts to distribute updates for the DDPG algorithm, (e.g. Popov et al., 2017) and more generally in this work we build on work of (Horgan et al., 2018) for implementing distributed actors.

Recently, Bellemare et al. (2017) showed that the distribution over returns, whose expectation is the value function, obeys a distributional Bellman equation. Although the idea of estimating a distribution over returns has been revisited before (Sobel, 1982; Morimura et al., 2010), Bellemare et al. demonstrated that this estimation alone was enough to achieve state-of-the-art results on the Atari 2600 benchmarks. Crucially, this technique achieves these gains by directly improving updates for the critic.

## 2 BACKGROUND

In this work we consider a standard reinforcement learning setting wherein an agent interacts with an environment in discrete time. At each timestep $t$ the agent makes observations $\mathbf{x}_t \in \mathcal{X}$, takes actions $\mathbf{a}_t \in \mathcal{A}$, and receives rewards $r(\mathbf{x}_t, \mathbf{a}_t) \in \mathbb{R}$. Although we will in general make no assumptions about the inputs $\mathcal{X}$, we will assume that the environments considered in this work have real-valued actions $\mathcal{A} = \mathbb{R}^d$.

In this standard setup, the agent's behavior is controlled by a policy $\pi : \mathcal{X} \to \mathcal{A}$ which maps each observation to an action. The state-action value function, which describes the expected return conditioned on first taking action $\mathbf{a} \in \mathcal{A}$ from state $\mathbf{x} \in \mathcal{X}$ and subsequently acting according to $\pi$, is defined as

$$Q_\pi(\mathbf{x}, \mathbf{a}) = \mathbb{E}\Big[ \sum_{t=0}^{\infty} \gamma^t r(\mathbf{x}_t, \mathbf{a}_t) \Big] \quad \text{where} \quad \begin{aligned} &\mathbf{x}_0 = \mathbf{x}, \ \mathbf{a}_0 = \mathbf{a}, \\ &\mathbf{x}_t \sim p(\cdot | \mathbf{x}_{t-1}, \mathbf{a}_{t-1}), \\ &\mathbf{a}_t = \pi(\mathbf{x}_t), \end{aligned} \tag{1}$$

and is commonly used to evaluate the quality of a policy. While it is possible to derive an updated policy directly from $Q_\pi$, such an approach typically requires maximizing this function

with respect to $\mathbf{a}$ and is made complicated by the continuous action space. Instead we will consider a parameterized policy $\pi_\theta$ and maximize the expected value of this policy by optimizing $J(\theta) = \mathbb{E}[Q_{\pi_\theta}(\mathbf{x}, \pi_\theta(\mathbf{x}))]$. By making use of the deterministic policy gradient theorem (Silver et al., 2014) one can write the gradient of this objective as

$$\nabla_\theta J(\theta) \approx \mathbb{E}_\rho \Big[ \nabla_\theta \pi_\theta(\mathbf{x}) \, \nabla_\mathbf{a} Q_{\pi_\theta}(\mathbf{x}, \mathbf{a}) \big|_{\mathbf{a}=\pi_\theta(\mathbf{x})} \Big], \tag{2}$$

where $\rho$ is the state-visitation distribution associated with some behavior policy. Note that by letting the behavior policy differ from $\pi$ we are able to empirically evaluate this gradient using data gathered off-policy.

While the exact gradient given by (2) assumes access to the true value function of the current policy, we can instead approximate this quantity with a parameterized critic $Q_w(\mathbf{x}, \mathbf{a})$. By introducing the Bellman operator

$$(\mathcal{T}_\pi Q)(\mathbf{x}, \mathbf{a}) = r(\mathbf{x}, \mathbf{a}) + \gamma \mathbb{E}\big[ Q(\mathbf{x}', \pi(\mathbf{x}')) \,\big|\, \mathbf{x}, \mathbf{a} \big], \tag{3}$$

whose expectation is taken with respect to the next state $\mathbf{x}'$, we can minimize the temporal difference (TD) error, i.e. the difference between the value function before and after applying the Bellman update. Typically the TD error will be evaluated under separate *target* policy and value networks, i.e. networks with separate parameters $(\theta', w')$, in order to stabilize learning. By taking the two-norm of this error we can write the resulting loss as

$$L(w) = \mathbb{E}_\rho \Big[ (Q_w(\mathbf{x}, \mathbf{a}) - (\mathcal{T}_{\pi_{\theta'}} Q_{w'})(\mathbf{x}, \mathbf{a}))^2 \Big]. \tag{4}$$

In practice we will periodically replace the target networks with copies of the current network weights. Finally, by training a neural network policy using the deterministic policy gradient in (2) and training a deep neural to minimize the TD error in (4) we obtain the Deep Deterministic Policy Gradient (DDPG) algorithm (Lillicrap et al., 2016). Here a sample-based approximation to these gradients is employed by using data gathered in some replay table.

## 3 DISTRIBUTED DISTRIBUTIONAL DDPG

The approach taken in this work starts from the DDPG algorithm and includes a number of enhancements. These extensions, which we will detail in this section, include a distributional critic update, the use of distributed parallel actors, $N$-step returns, and prioritization of the experience replay.

First, and perhaps most crucially, we consider the inclusion of a distributional critic as introduced in Bellemare et al. (2017). In order to introduce the distributional update we first revisit (1) in terms of the return as a random variable $Z_\pi$, such that $Q_\pi(\mathbf{x}, \mathbf{a}) = \mathbb{E}\, Z_\pi(\mathbf{x}, \mathbf{a})$. The distributional Bellman operator can be defined as

$$(\mathcal{T}_\pi Z)(\mathbf{x}, \mathbf{a}) = r(\mathbf{x}, \mathbf{a}) + \gamma \mathbb{E}\big[ Z(\mathbf{x}', \pi(\mathbf{x}')) \,\big|\, \mathbf{x}, \mathbf{a} \big], \tag{5}$$

where equality is with respect to the probability law of the random variables; note that this expectation is taken with respect to distribution of $Z$ as well as the transition dynamics.

While the definition of this operator looks very similar to the canonical Bellman operator defined in (3), it differs in the types of functions it acts on. The distributional variant takes functions which map from state-action pairs to distributions, and returns a function of the same form. In order to use this function within the context of the actor-critic architecture introduced above, we must parameterize this distribution and define a loss similar to that of Equation 4. We will write the loss as

$$L(w) = \mathbb{E}_\rho \Big[ d(\mathcal{T}_{\pi_{\theta'}} Z_{w'}(\mathbf{x}, \mathbf{a}), Z_w(\mathbf{x}, \mathbf{a})) \Big] \tag{6}$$

for some metric $d$ that measures the distance between two distributions. Two components that can have a significant impact on the performance of this algorithm are the specific parameterization used for $Z_w$ and the metric $d$ used to measure the distributional TD error. In both cases we will give further details in Appendix A; in the experiments that follow we will use the Categorical distribution detailed in that section.

We can complete this distributional policy gradient algorithm by including the action-value distribution inside the actor update from Equation 2. This is done by taking the expectation with respect to the action-value distribution, i.e.

$$\nabla_\theta J(\theta) \approx \mathbb{E}_\rho \Big[ \nabla_\theta \pi_\theta(\mathbf{x}) \, \nabla_\mathbf{a} Q_w(\mathbf{x}, \mathbf{a}) \big|_{\mathbf{a}=\pi_\theta(\mathbf{x})} \Big],$$
$$= \mathbb{E}_\rho \Big[ \nabla_\theta \pi_\theta(\mathbf{x}) \, \mathbb{E}[\nabla_\mathbf{a} Z_w(\mathbf{x}, \mathbf{a})] \big|_{\mathbf{a}=\pi_\theta(\mathbf{x})} \Big]. \tag{7}$$

---

**Algorithm 1** D4PG

---

**Input:** batch size $M$, trajectory length $N$, number of actors $K$, replay size $R$, exploration constant $\epsilon$, initial learning rates $\alpha_0$ and $\beta_0$

1: Initialize network weights $(\theta, w)$ at random
2: Initialize target weights $(\theta', w') \leftarrow (\theta, w)$
3: Launch $K$ actors and replicate network weights $(\theta, w)$ to each actor
4: **for** $t = 1, \dots, T$ **do**
5:     Sample $M$ transitions $(\mathbf{x}_{i:i+N}, \mathbf{a}_{i:i+N-1}, r_{i:i+N-1})$ of length $N$ from replay with priority $p_i$

6:     Construct the target distributions $Y_i = \sum_{n=0}^{N-1} \gamma^n r_{i+n} + \gamma^N Z_{w'}(\mathbf{x}_{i+N}, \pi_{\theta'}(\mathbf{x}_{i+N}))$
7:     Compute the actor and critic updates

$$\delta_w = \frac{1}{M} \sum_i \nabla_w (Rp_i)^{-1} d(Y_i, Z_w(\mathbf{x}_i, \mathbf{a}_i))$$

$$\delta_\theta = \frac{1}{M} \sum_i \nabla_\theta \pi_\theta(\mathbf{x}_i) \, \mathbb{E}[\nabla_\mathbf{a} Z_w(\mathbf{x}_i, \mathbf{a})]\big|_{\mathbf{a} = \pi_\theta(\mathbf{x}_i)}$$

8:     Update network parameters $\theta \leftarrow \theta + \alpha_t\, \delta_\theta$, $w \leftarrow w + \beta_t\, \delta_w$
9:     If $t = 0 \mod t_{\text{target}}$, update the target networks $(\theta', w') \leftarrow (\theta, w)$
10:     If $t = 0 \mod t_{\text{actors}}$, replicate network weights to the actors
11: **end for**
12: **return** policy parameters $\theta$

---

**Actor**

---

1: **repeat**
2:     Sample action $\mathbf{a} = \pi_\theta(\mathbf{x}) + \epsilon \mathcal{N}(0, 1)$
3:     Execute action $\mathbf{a}$, observe reward $r$ and state $\mathbf{x}'$
4:     Store $(\mathbf{x}, \mathbf{a}, r, \mathbf{x}')$ in replay
5: **until** learner finishes

---

As before, this update can be empirically evaluated by replacing the outer expectation with a sample-based approximation.

Next, we consider a modification to the DDPG update which utilizes $N$-step returns when estimating the TD error. This can be seen as replacing the Bellman operator with an $N$-step variant

$$(\mathcal{T}_\pi^N Q)(\mathbf{x}_0, \mathbf{a}_0) = r(\mathbf{x}_0, \mathbf{a}_0) + \mathbb{E}\Big[ \sum_{n=1}^{N-1} \gamma^n r(\mathbf{x}_n, \mathbf{a}_n) + \gamma^N Q(\mathbf{x}_N, \pi(\mathbf{x}_N)) \,\big|\, \mathbf{x}_0, \mathbf{a}_0 \Big] \tag{8}$$

where the expectation is with respect to the $N$-step transition dynamics. Although not used by Lillicrap et al. (2016), $N$-step returns are widely used in the context of many policy gradient algorithms (e.g. Mnih et al., 2016) as well as Q-learning variants (Hessel et al., 2017). This modification can be applied analogously to the distributional Bellman operator in order to make use of it when updating the distributional critic.

Finally, we also modify the standard training procedure in order to distribute the process of gathering experience. Note from Equations (2,4) that the actor and critic updates rely entirely on sampling from some state-visitation distribution $\rho$. We can parallelize this process by using $K$ independent actors, each writing to the same replay table. A learner process can then sample from some replay table of size $R$ and perform the necessary network updates using this data. Additionally sampling can be implemented using non-uniform priorities $p_i$ as in Schaul et al. (2016). Note that this requires the use of importance sampling, implemented by weighting the critic update by a factor of $1/Rp_i$. We implement this procedure using the ApeX framework (Horgan et al., 2018) and refer the reader there for more details.

Algorithm pseudocode for the D4PG algorithm which includes all the above-mentioned modifications can be found in Algorithm 1. Here the actor and critic parameters are updated using stochastic gradient descent with learning rates, $\alpha_t$ and $\beta_t$ respectively, which are adjusted online using ADAM (Kingma & Ba, 2015). While this pseudocode focuses on the learning process, also shown is pseudocode for actor processes which in parallel fill the replay table with data.

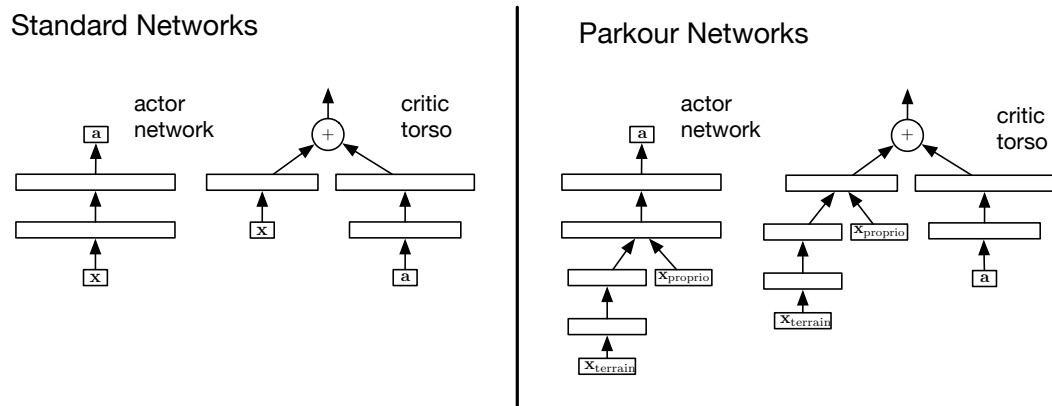

Figure 1: Architectural variants used for each domain. The left-most set illustrates the actor network and critic torso used for the standard control and manipulation domains. The full critic architecture is completed by feeding the output of the critic torso into a relevant distribution, e.g. the categorical distribution, as defined in Section A. The right half of the figure similarly illustrates the architecture used by the parkour domains.

## 4 RESULTS

In this section we describe the performance of the D4PG algorithm across a variety of continuous control tasks. To do so, in each environment we run our learning procedure and periodically snapshot the policy in order to test it without exploration noise. We will primarily be interested in the performance as a function of wall clock time, however we will also examine the data efficiency. Most interestingly, from a scientific perspective, we also perform a number of ablations which individually remove components of the D4PG algorithm in order to determine their specific contributions.

First, we experiment with and without distributional updates. In this setting we focus on use of a categorical distribution as we found in preliminary experiments that the use of a mixture of Gaussians performed worse and was less stable with respect to hyperparameter values across different tasks; a selection of these runs can be found in Appendix C. Across all tasks—except for one which we will introduce later—we use 51 atoms for the categorical distribution. In what follows we will refer to non-distributional variants of this algorithm as Distributed DDPG (D3PG).

Next, we consider prioritized and non-prioritized versions of these algorithm variants. For the non-prioritized variants, transitions are sampled from replay uniformly. For prioritized variants we use the absolute TD-error to sample from replay in the case of D3PG, and for D4PG we use the absolute distributional TD-error as described in Section A. We also vary the trajectory length $N \in \{1, 5\}$.

In all experiments we use a replay table of size $R = 1 \times 10^6$ and only consider behavior policies which add fixed Gaussian noise $\epsilon \mathcal{N}(0, 1)$ to the current online policy; in all experiments we use a value of $\epsilon = 0.3$. We experimented with correlated noise drawn from an Ornstein-Uhlenbeck process, as suggested by (Lillicrap et al., 2016), however we found this was unnecessary and did not add to performance. For all algorithms we initialize the learning rates for both actor and critic updates to the same value. In the next section we will present a suite of simple control problems for which this value corresponds to $\alpha_0 = \beta_0 = 1 \times 10^{-4}$; for the following, harder problems we set this to a smaller value of $\alpha_0 = \beta_0 = 5 \times 10^{-5}$. Similarly for the control suite we utilize a batch size of $M = 256$ and for all subsequent problems we will increase this to $M = 512$.

### 4.1 STANDARD CONTROL SUITE

We first consider evaluating performance on a number of simple, physical control tasks by utilizing a suite of benchmark tasks (Tassa et al., 2018) developed in the MuJoCo physics simulator (Todorov et al., 2012). Each task is run for exactly 1000 steps and provides either an immediate dense reward $r_t \in [0, 1]$ or sparse reward $r_t \in \{0, 1\}$ depending on the particular task. For each domain, the inputs presented to the agent consist of reasonably low-dimensional observations, many consisting of phys-

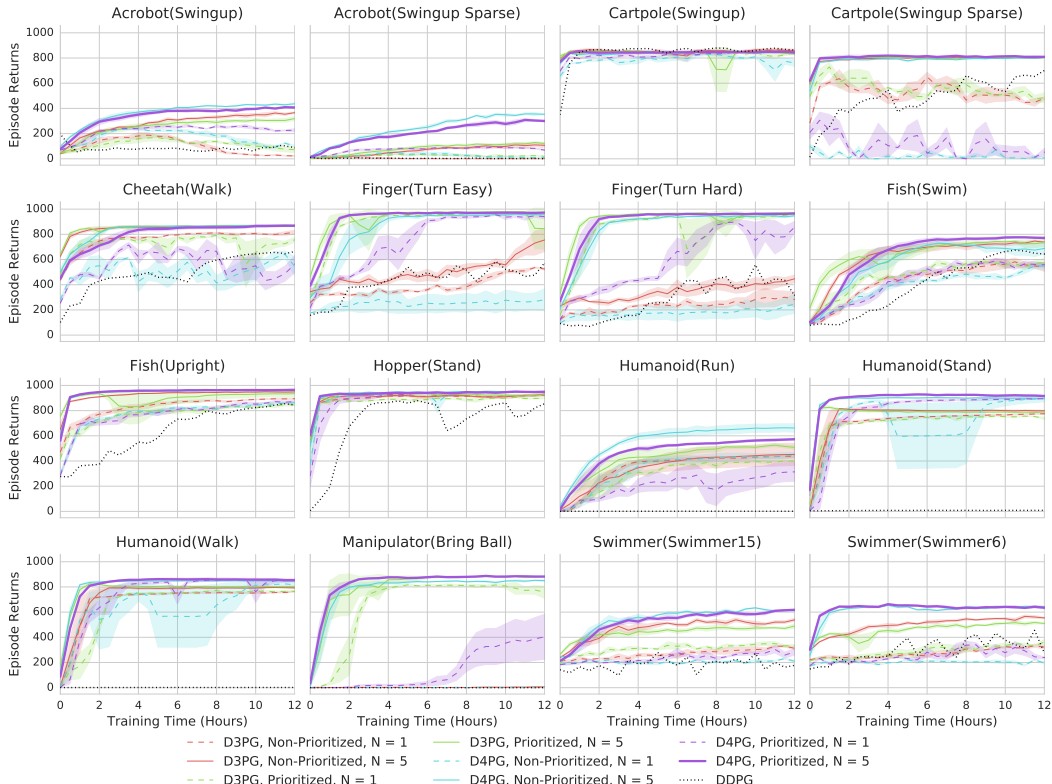

Figure 2: Experimental results across domains in the control suite.

ical state, joint angles, etc. These observations range between 6 and 60 dimensions, however note that the difficulty of the task is not immediately associated with its dimensionality. For example the acrobot is one of the lowest dimensional tasks in this suite which, due to its level of controllability, can prove much more difficult to learn than other, higher dimensional tasks. For an illustration of these domains see Figure 9; see Appendix D for more details.

For algorithms in these experiments we consider actor and critic architectures of the form given in Figure 1 and for each experiment we use $K = 32$ actors. Figure 2 shows the performance of D4PG and its various ablations across the entire suite of control tasks. This set of plots is quite busy, however it serves as a broad set of tasks with which we can obtain a general idea of the algorithms performance. Later experiments on harder domains look more closely at the difference between algorithms. Here we also compare against the canonical (non-distributed) DDPG algorithm as a baseline, shown as a dotted black line. This removes all the enhancements proposed in this paper, and we can see that except on the simplest domain, Cartpole (Swingup), it performs worse than all other methods. This performance disparity worsens as we increase the difficulty of tasks, and hence for further experiments we will drop this line from the plot.

Next, across all tasks we see that the best performance is obtained by the full D4PG algorithm (shown in purple and bold). Here we see that the longer unroll length of $N = 5$ is uniformly better (we show these as solid lines), and in particular we sometimes see for both D3PG and D4PG that an unroll length of $N = 1$ (shown as dashed lines) can occasionally result in instability. This is especially apparent in the Cheetah (Walk) and Cartpole (Swingup Sparse) tasks.

The next biggest gain is arguably due to the inclusion of the distributional critic update, where it is particularly helpful on the hardest tasks e.g. Humanoid (Run) and Acrobot. The manipulator is also quite difficult among this suite of tasks, and here we see that the inclusion of the distributional update does not help as much as in other tasks, although note that here the D3PG and D4PG variants obtain approximately the same performance. As far as the use of prioritization is concerned, it does not appear to contribute significantly to the performance of D4PG. This is not the case for D3PG, however, which on many tasks is helped significantly by the inclusion of prioritization.

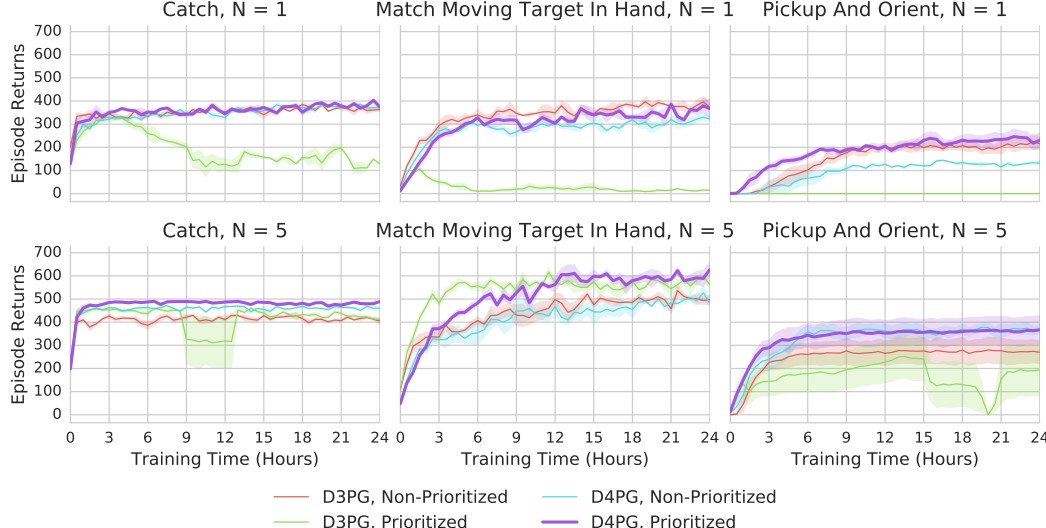

Figure 3: Experimental results for tasks in the manipulation domain.

## 4.2 MANIPULATION

Next, we consider a set of tasks designed to highlight the ability of the D4PG agent to learn dexterous manipulation. Tasks of this form can prove difficult for many reasons, most notably the higher dimensionality of the control task, intermittent contact dynamics, and potential under-actuation of the manipulator.

Here we use a simulated hand model implemented within MuJoCo, consisting of 13 actuators which control 22 degrees of freedom. For these experiments the wrist site is attached to a fixed location in space, about which it is allowed to rotate axially. In particular this allows the hand to pick up objects, rotate into a palm-up position, and manipulate them. We first consider a task in which a cylinder is dropped onto the hand from a random height, and the goal of the task is to catch the falling cylinder. The next task requires the agent to pick up an object from the tabletop and then maneuver it to a target position and orientation. The final task is one wherein a broad cylinder must be rotated in-hand in order to match a target orientation. See Appendix E for further details regarding both the model and the tasks. For these tasks we use the same network architectures as in the previous section as well as $K = 64$ actors.

In Figure 3 we again compare the D4PG algorithm against ablations of its constituent components. Here we split the algorithms between $N = 1$ in the top row and $N = 5$ in the bottom row, and in particular we can see that across all algorithms $N = 5$ is uniformly better. For all tasks, the full D4PG algorithm performs either at the same level or better than other ablations; this is particularly apparent in the $N = 5$ case. Overall the use of priorization never seems to harm D4PG, however it does appear to be of limited additional value. Interestingly this is not necessarily the case with the D3PG variant (i.e. without distributional updates). Here we can see that prioritization sometimes harms the performance of D3PG, and this is very readily seen in the $N = 1$ case where the algorithm can either become unstable, or in the case of the Pickup and Orient task it completely fails to learn.

## 4.3 PARKOUR

Finally, we consider the *parkour* domain introduced by (Heess et al., 2017). In this setting the agent controls a simplified robotic walker which is rewarded for forward movement, but is impeded by a number of randomly sampled obstacles; see Figure 4 for a visualization and refer to the earlier work for further details. The first of our experiments considers a two-dimensional walker, i.e. a domain in which the walker is allowed to move horizontally and vertically, but is constrained to a fixed depth position. In this domain the obstacles presented to the agent include gaps in the floor surface, barriers it must jump over, and platforms that it can either run over or underneath. The agent is presented with proprioceptive observations $x_{proprio} \in \mathbb{R}^{19}$ corresponding to the angles of its limbs and other functions of these quantities. It is also given access to observations $x_{terrain} \in \mathbb{R}^{101}$

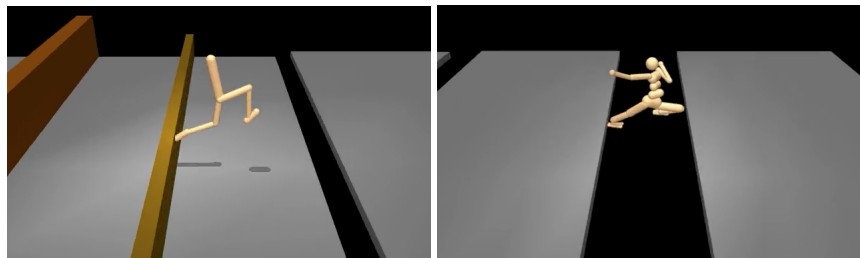

Figure 4: Example frames taken from trained agents running in the two parkour domains.

which includes features such as a depth map of the upcoming terrain, etc. In order to accommodate these inputs we utilize a network architecture as specified in Figure 1. In particular we make use of a stack of feed-forward layers which process the terrain information to reduce it to a smaller number of hidden units before concatenating with the proprioceptive information for further processing. The actions in this domain take the form of torque controls $\mathbf{a} \in \mathbb{R}^6$.

In order to examine the performance of the D4PG algorithm in this setting we consider the ablations of the previous sections and we have further introduced a PPO baseline as utilized in the earlier paper of (Heess et al., 2017). For all algorithms, including PPO, we use $K = 64$ actors. These results are shown in Figure 5 in the top row. As before we examine the performance separately for $N = 1$ and $N = 5$, and again we see that the higher unroll length results in better performance. Note that we show the PPO baseline on both plots for consistency, but in both plots this is the same algorithm, with settings proposed in the earlier paper and unrolls of length 50.

Here we again see a clear delineation and clear gains for each of the other algorithm components. The biggest gain comes from the inclusion of the distributional update, which we can see by comparing the non-prioritized D3PG/D4PG variants. We see marginal benefit to using prioritization for D3PG, but this gain disappears when we consider the distributional update. Finally, we can see when comparing to the PPO baseline that this algorithm compares favorably to D3PG in the case of $N = 1$, however is outperformed by D4PG; when $N = 5$ all algorithms outperform PPO.

Next, in the plots shown in Figure 5 on the bottom row we also consider the performance not just in terms of training time, but also in terms of the sample complexity. In order to do so we plot the performance of each algorithm versus the number of actor steps, i.e. the quantity of transitions collected. This is perhaps more favorable to PPO, as the parallel actors considered in this work are not necessarily tuned for sample efficiency. Here we see that PPO is able to out-perform the non-prioritized version of D3PG, and early on in training is favorable compared to the prioritized version, although this trails off. However, we still see significant performance gains by utilizing the distributional updates, both in a prioritized and non-prioritized setting. Interestingly we see that the use of prioritization does not gain much, if any over the non-prioritized D4PG version. Early in the trajectory for $N = 5$, in fact, we see that the non-prioritized D4PG exhibits better performance, however later these performance curves level out. With respect to wall-clock time these small differences may be due to small latencies in the scheduling of different runs, as we see that this difference is less for the plot with respect to actor steps.

Finally we consider a humanoid walker which is able to move in all three dimensions. The obstacles in this domain consist of gaps in the floor, barriers that must be jumped over, and walls with gaps that allow the agent to run through. For this experiment we utilize the same network architecture as in the previous experiment, except now the observations are of size $\mathbf{x}_{\text{proprio}} \in \mathbb{R}^{79}$ and $\mathbf{x}_{\text{terrain}} \in \mathbb{R}^{461}$. Again actions are torque controls, but in 21 dimensions. In this task we also increased the number of atoms for the categorical distribution from 51 to 101. This change increases the level of resolution for the distribution in order to keep the resolution roughly consistent with other tasks. This is a much higher dimensional problem than the previous parkour task with a significantly more difficult control task: the walker is more unstable and there are many more ways for the agent to fail than in the previous experiment. The results for this particular domain are displayed in Figure 6, and here we concentrate on performance as a function of wall-clock time, restricted to the previously best performing roll-out length of $N = 5$. In this setting we see a clear delineation between first the PPO results which are the poorest performing, the D3PG results where the prioritized version has a slight edge, and finally the D4PG results. Interestingly for D4PG we again see as in the two-dimensional walker case, the use of prioritization seems to have no benefit, with both versions have

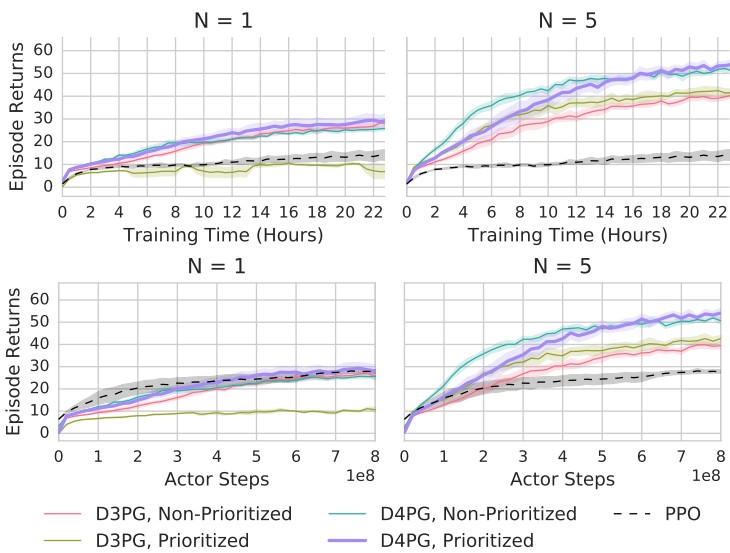

Figure 5: Experimental results for the two-dimensional (walker) parkour domain when compared first versus wall-clock time (top) and versus actor steps (bottom).

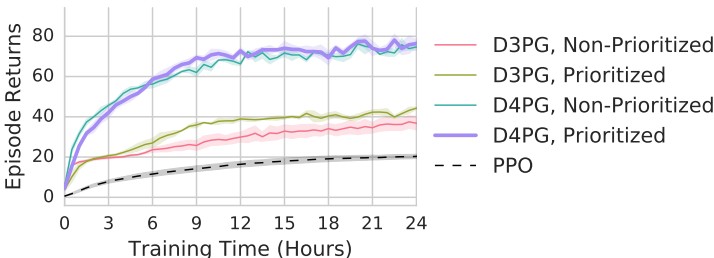

Figure 6: Experimental results for the three-dimensional (humanoid) parkour domain.

almost identical performance curves; in fact the performance here is perhaps even closer than that of the previous set of experiments.

## 5 DISCUSSION

In this work we introduced the D4PG, or Distributed Distributional DDPG, algorithm. Our main contributions include the inclusion of a distributional updates to the DDPG algorithm, combined with the use of multiple distributed workers all writing into the same replay table. We also consider a number of other, smaller changes to the algorithm. All of these simple modifications contribute to the overall performance of the D4PG algorithm; the biggest performance gain of these simple changes is arguably the use of $N$-step returns. Interestingly we found that the use of priority was less crucial to the overall D4PG algorithm especially on harder problems. While the use of prioritization was definitely able to increase the performance of the D3PG algorithm, we found that it can also lead to unstable updates. This was most apparent in the manipulation tasks.

Finally, as our results can attest, the D4PG algorithm is capable of state-of-the-art performance on a number of very difficult continuous control problems.

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

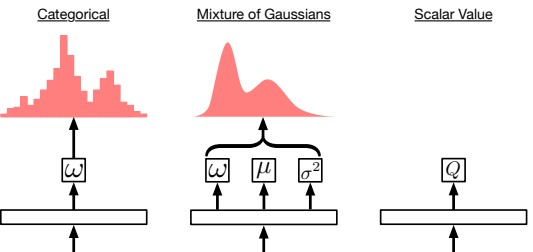

Figure 7: Output layers corresponding to different distribution parameterizations. From left to right these include the Categorical, Mixture of Gaussians, and finally the standard scalar value function.

## A DISTRIBUTIONS AND LOSSES

In this section we consider two potential parameterized distributions for D4PG. Parameterized distributions, in this framework, are implemented as a neural network layer mapping the output of the critic torso (see Figure 1) to the parameters of a given distribution (e.g. mean and variance). In what follows we will detail the distributions and their corresponding losses.

**Categorical** Following Bellemare et al. (2017), we first consider the categorical parameterization, a layer whose parameters are the logits $\omega_i$ of a discrete-valued distribution defined over a fixed set of atoms $z_i$. This distribution has hyperparameters for the number of atoms $\ell$, and the bounds on the support $(V_{\min}, V_{\max})$. Given these, $\Delta = \frac{V_{\max} - V_{\min}}{\ell - 1}$ corresponds to the distance between atoms, and $z_i = V_{\min} + i\Delta$ gives the location of each atom. We can then define the action-value distribution as

$$Z = z_i \quad \text{w.p.} \quad p_i \propto \exp\{\omega_i\}. \tag{9}$$

Observe that this distributional layer simply corresponds to a linear layer from the critic torso to the logits $\omega$, followed by a softmax activation (see Figure 7, left).

However, this distribution is not closed under the Bellman operator defined earlier, due to the fact that adding and scaling these values will no longer lie on the support defined by the atoms. This support is explicitly defined by the $(V_{\min}, V_{\max})$ hyperparameters. As a result we instead use a projected version of the distributional Bellman operator (Bellemare et al., 2017); see Appendix B for more details. Letting $p'$ be the probabilities of the projected distributional Bellman operator $\Phi\mathcal{T}_\pi$ applied to some target distribution $Z_{\text{target}}$, we can write the loss in terms of the cross-entropy

$$d(\Phi\mathcal{T}_\pi Z_{\text{target}}, Z) = \sum_{i=0}^{\ell-1} p'_i \frac{\exp\{\omega_i\}}{\sum_j \exp\{\omega_j\}}. \tag{10}$$

**Mixture of Gaussians** We can also consider parameterizing the action-value distribution using a mixture of Gaussians; here the random variable $Z$ has density given by

$$p(z) \propto \sum_{i=0}^{\ell-1} \omega_i \, \mathcal{N}(z \,|\, \mu_i, \sigma_i^2). \tag{11}$$

Thus, the distribution layer maps, through a linear layer, from the critic torso to the mixture weight $\omega_i$, mean $\mu_i$, and variance $\sigma_i^2$ for each mixture component $0 \leqslant i \leqslant \ell - 1$ (see Figure 7, center). We can then specify a loss corresponding to the cross-entropy portion of the KL divergence between two distributions. Given a sample transition $(\mathbf{x}, \mathbf{a}, r, \mathbf{x}')$ we can take samples from the target density $z_j \sim p_{\text{target}}$ and approximate the cross-entropy term using

$$d(\mathcal{T}_\pi Z_{\text{target}}, Z) \approx \sum_j \log p(r + \gamma z_j). \tag{12}$$

## B CATEGORICAL PROJECTION OPERATOR

The categorical parameterized distribution has finite support. Thus, the result of applying the distributional Bellman equation will generally not coincide with this support. Therefore, some projection

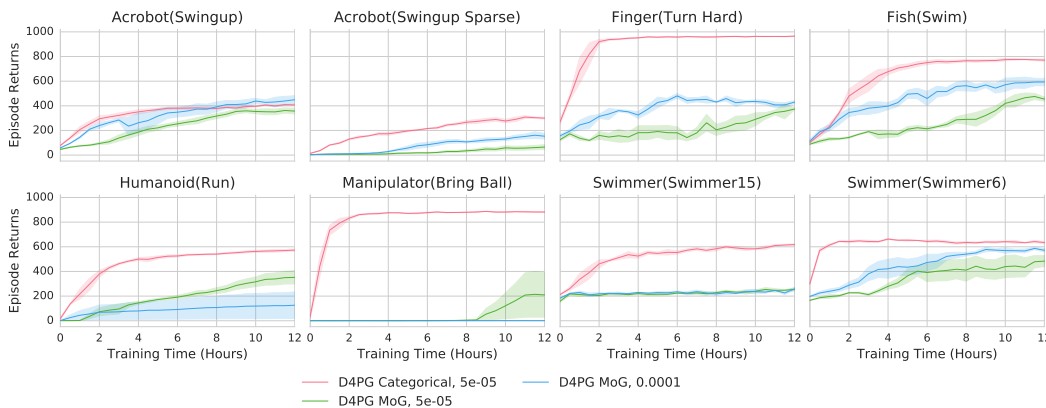

Figure 8: Results for using a mixture of Gaussians distribution on select control suite tasks. Shown are two learning rates as denoted in the legends as well as Categorical.

step is required before minimizing the cross-entropy. The categorical projection of Bellemare et al. (2017) is given by $(\Phi p)_i = \sum_{j=0}^{\ell-1} h_{z_i}(z_j) p_j, \ \forall i$, where $h$ is a piecewise linear 'hat' function,

$$
h_{z_i}(z) = \begin{cases} 1 & z \leqslant V_{\min} \text{ and } i = 0, \\ \frac{z - z_{i-1}}{z_i - z_{i-1}} & \text{for } z_{i-1} \leqslant z \leqslant z_i, \\ \frac{z_{i+1} - z}{z_{i+1} - z_i} & \text{for } z_i \leqslant z \leqslant z_{i+1}, \\ 1 & z \geqslant V_{\max} \text{ and } i = \ell - 1. \end{cases} \tag{13}
$$

## C  MIXTURES OF GAUSSIANS CONTROL SUITE RESULTS

In Figure 8 we display results of running D4PG on a selection of control suite tasks using a mixture of Gaussians output distribution for two choices of learning rates. Here the distributional TD loss is minimized using the sample-based KL introduced earlier. While this is definitely a technique that is worth further exploration, we found in initial experiments that this choice of distribution underperformed the Categorical distribution by a fair margin. This lends further credence to the choice of distribution made in (Bellemare et al., 2017).

## D  CONTROL SUITE DETAILS

In this section we provide further details for the control suite domains. In particular see Figure 9 for images of the control suite tasks. The physics state $\mathcal{S}$, action $\mathcal{A}$, and observation $\mathcal{X}$ dimensionalities for each task are provided in Table 1.

## E  MANIPULATION DETAILS

For the dexterous manipulation tasks we used a simulated model of the Johns Hopkins Modular Prosthetic Limb hand (Johannes et al., 2011) implemented in MuJoCo (Kumar & Todorov, 2015). This anthropomorphic hand has a total of 22 degrees of freedom (19 in the fingers, 3 in the wrist), which are driven by a set of 13 position actuators (PD-controllers). The underactuation of the hand is due to coupling between some of the finger joints. For these experiments the wrist was positioned in a fixed location above a table, such that rotation and flexion about the wrist joints allowed the hand to pick up objects from the table, rotate into a palm-up position, and then manipulate them.

We focused on a set of three tasks where the agent must learn to manipulate a cylindrical object (Figure 10). In each of these tasks, the observations contain the positions and velocities of all of the joints in the hand, the current position targets for the actuators in the hand, the position and quaternion of the object being manipulated, and its translational and rotational velocities. The

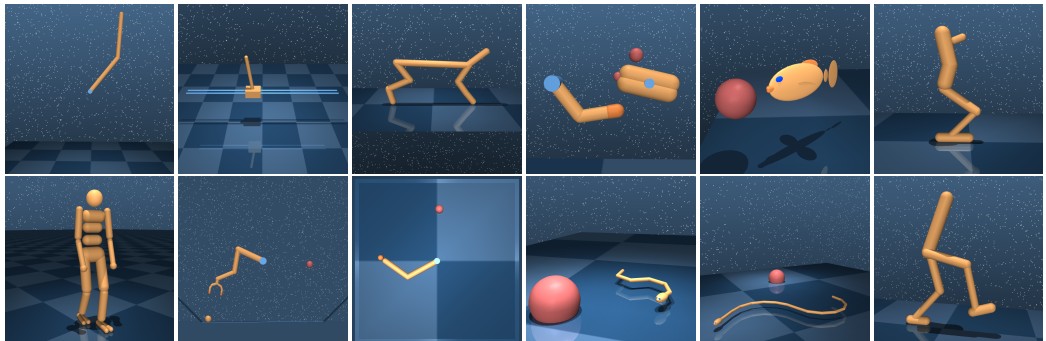

Figure 9: Control Suite domains used for benchmarking. *Top*: acrobot, cartpole, cheetah, finger, fish, hopper. *Bottom*: humanoid, manipulator, pendulum, reacher, swimmer6, swimmer15, walker.

| Domain | Task | $\|\mathcal{A}\|$ | $\|\mathcal{S}\|$ | $\|\mathcal{X}\|$ |
|---|---|---|---|---|
| acrobot | swingup
swingup_sparse | 1 | 4 | 6 |
| cartpole | swingup
swingup_sparse | 1 | 4 | 5 |
| cheetah | walk | 6 | 18 | 17 |
| finger | turn_easy
turn_hard | 2 | 6 | 12 |
| fish | upright
swim | 5 | 27 | 24 |
| hopper | stand | 4 | 14 | 15 |
| humanoid | stand
walk
run | 21 | 55 | 67 |
| manipulator | bring_ball | 2 | 22 | 37 |
| swimmer | swimmer6
swimmer15 | 5
14 | 16
34 | 25
61 |

Table 1: Domains and tasks in the Control Suite.

| | | | | Task | |
|---|---|---|---|---|---|
| | | Size | Catch | Pick-up-and-orient | Rotate-in-hand |
| Hand | joint positions | 22 | ✓ | ✓ | ✓ |
| | joint velocities | 22 | ✓ | ✓ | ✓ |
| | actuator targets | 13 | ✓ | ✓ | ✓ |
| Object | position | 3 | ✓ | ✓ | ✓ |
| | quaternion | 4 | ✓ | ✓ | ✓ |
| | velocity | 6 | ✓ | ✓ | ✓ |
| Target | position | 3 | – | ✓ | – |
| | quaternion | 4 | – | ✓ | – |
| | $\sin_z$, $\cos_z$ | 2 | – | – | ✓ |
| Total | | | 70 | 77 | 72 |

Table 2: Observation components given in each of the manipulation tasks, and their corresponding dimensionalities. Here $\sin_z$, $\cos_z$ refers to the sine and cosine of the target frame's angle of rotation about the $z$-axis.

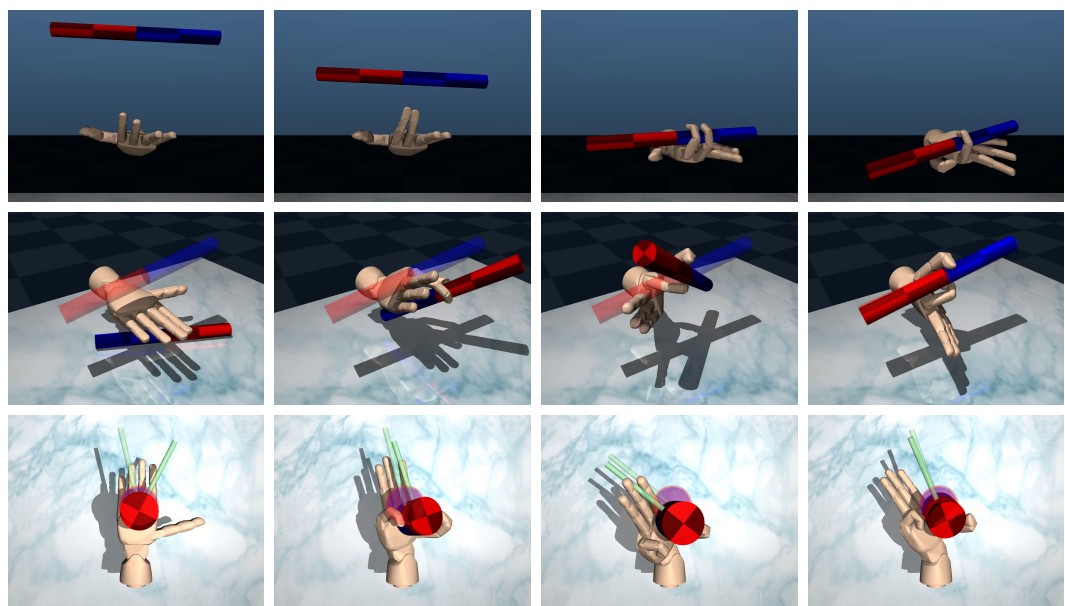

Figure 10: Sequences of frames illustrating the dexterous manipulation tasks we attempt to solve using D4PG. *Top to bottom*: 'catch', 'pick-up-and-orient', 'rotate-in-hand'. The translucent objects shown in 'pick-up-and-orient' and 'rotate-in-hand' represent the goal states.

observations given in each task are summarized in Table 2. The agent's actions are increments applied to the position targets for the actuators.

In the 'catch' task the agent must learn to catch a falling object before it strikes the table below. The position, height, and orientation of the object are randomly initialized at the start of each episode. The reward is given by

$$r = \psi(\text{palm}_{\text{height}} - \text{obj}_{\text{height}}; c, m) \tag{14}$$

where $\psi(\epsilon; c, m)$ is a soft indicator function similar to one described by Hafner & Riedmiller (2011)

$$\psi(\epsilon; c, m) = \begin{cases} 1 - \tanh(\frac{w}{m}\epsilon)^2 & \text{if } \epsilon > c, \\ 1 & \text{otherwise.} \end{cases} \tag{15}$$

Here $w = \tanh^{-1}(\sqrt{0.95})$, and the tolerance $c$ and margin $m$ parameters are $0\,\text{cm}$ and $5\,\text{cm}$ respectively. Contact between the object and the table causes the current episode to terminate immediately with no reward, otherwise it will continue until a 500 step limit is reached.

In the 'pick-up-and-orient' task, the agent must pick up a cylindrical object from the table and maneuver it into a target position and orientation. Both the initial position and orientation of the object, and the position and orientation of the target are randomized between episodes. The reward function consists of two additive components that depend on the distance from the object to the target position, and on the angle between the $z$-axes of the object and target body frames

$$r = 0.5\psi(||\text{obj}_{\text{pos}} - \text{obj}_{\text{pos}}^{\text{target}}||_2; c_{\text{pos}}, m_{\text{pos}})(1 + \psi(\cos^{-1}(\text{obj}_{\text{zaxis}} \cdot \text{obj}_{\text{zaxis}}^{\text{target}}); c_{\text{ori}}, m_{\text{ori}})) \tag{16}$$

where $c_{\text{pos}}=1\,\text{cm}$, $m_{\text{pos}}=5\,\text{cm}$, $c_{\text{ori}}=5°$, $m_{\text{ori}}=10°$. Note that the distance-dependent component of the reward multiplicatively gates the orientation component. This helps to encourage the agent to pick up the object before attempting to orient it to match the target. Each episode has a fixed duration of 500 steps.

Finally, in the 'rotate-in-hand' task the agent begins with a broad cylinder in its palm, and must rotate it axially in order to match a moving target. This requires dynamically forming and breaking

contacts with the object being manipulated. The target angle is initialized uniformly, and then incremented on each time step using temporally correlated noise drawn from an Ornstein-Uhlenbeck process ($\sigma$=0.025°, $\theta$=0.01; Uhlenbeck & Ornstein 1930). The reward consists of two multiplicative components

$$r = \psi(\cos^{-1}(\mathrm{obj}_{\mathrm{yaxis||xy}}, \mathrm{obj}_{\mathrm{yaxis||xy}}^{\mathrm{target}})); c_{\mathrm{rot}}, m_{\mathrm{rot}})\psi(\cos^{-1}(\mathrm{obj}_{\mathrm{zaxis}}, \mathrm{obj}_{\mathrm{zaxis}}^{\mathrm{target}})); c_{\mathrm{ori}}, m_{\mathrm{ori}}) \quad (17)$$

where $c_{\mathrm{rot}}$=5°, $m_{\mathrm{rot}}$=40°, $c_{\mathrm{ori}}$=45°, $m_{\mathrm{ori}}$=45°, and $||$xy denotes projection onto the global $xy$ plane. The first component provides an incentive to match the axial rotation of the target, and the second component penalizes the agent for allowing the orientation of the cylinder's long axis to deviate too far from that of the target. The maximum episode duration is 1000 steps, with early termination if the object makes contact with the table.

