# OpenReview forum: "Distributed Distributional Deterministic Policy Gradients"
_ICLR.cc/2018/Conference — Accept (Poster)_

### Official Review · AnonReviewer2 · 2017-11-27
**Thorough investigation of distributional policy gradients for continuous problems**

**Rating:** 9
**Confidence:** 4

**Review:**

A DeepRL algorithm is presented that represents distributions over Q values, as applied to DDPG,
and in conjunction with distributed evaluation across multiple actors, prioritized experience replay, and
N-step look-aheads. The algorithm is called Distributed Distributional Deep Deterministic Policy Gradient algorithm, D4PG.
SOTA results are generated for a number of challenging continuous domain learning problems,
as compared to benchmarks that include DDPG and PPO, in terms of wall-clock time, and also (most often) in terms
of sample efficiency.

pros/cons
+ the paper provides a thorough investigation of the distributional approach, as applied to difficult continuous
  action problems, and in conjunction with a set of other improvements (with ablation tests)
- the story is a bit mixed in terms of the benefits, as compared to the non-distributional approach, D3PG
- it is not clear which of the baselines are covered in detail in the cited paper:
  "Anonymous. Distributed prioritized experience replay. In submission, 2017.",
   i.e., should readers assume that D3PG already exists and is attributable to this other submission?

Overall, I believe that the community will find this to be interesting work.

Is a video of the results available?

It seems that the distributional model often does not make much of a difference,
as compared to D3PG non-prioritized.  However, sometimes it does make a big difference, i.e., 3D parkour; acrobot.
Do the examples where it yields the largest payoff share a particular characteristic?

The benefit of the distributional models is quite different between the 1-step and 5-step versions. Any ideas why?

Occasionally, D4PG with N=1 fails very badly, e.g., fish, manipulator (bring ball), swimmer.
Why would that be? Shouldn't it do at least as well as D3PG in general?

How many atoms are used for the categorical representation?
As many as [Bellemare et al.], i.e., 51 ?
How much "resolution" is necessary here in order to gain most of the benefits of the distributional representation?

As far as I understand, V_min and V_max are not the global values, but are specific to the current distribution.
Hence the need for the projection. Is that correct?

Would increasing the exploration noise result in a larger benefit for the distributional approach?

Figure 2: DDPG performs suprisingly poorly in most examples. Any comments on this,
or is DDPG best avoided in normal circumstances for continuous problems? :-)

Is the humanoid stand so easy because of large (or unlimited) torque limits?

The wall-clock times are for a cluster with K=32 cores for Figure 1?

"we utilize a network architecture as specified in Figure 1 which processes the terrain info in order to reduce its dimensionality"
Figure 1 provides no information about the reduced dimensionality of the terrain representation, unless I am somehow failing to see this.

"the full critic architecture is completed by attaching a critic head as defined in Section A"
I could find no further documenation in the paper with regard to the "head" or a separate critic for the "head".
It is not clear to me why multiple critics are needed.

Do you have an intuition as to why prioritized replay might be reducing performance in many cases?

---

> ### Author Response · Authors · 2017-12-21
> **Response to AnonReviewer2**
>
> Thank you!
>
> As to the baselines, we use the same framework for distributing computation and prioritization as in the cited paper (Anonymous. Distributed prioritized experience replay, also submitted to ICLR). However this other work focuses primarily on discrete-action tasks.
>
> Videos of the parkour performance can be found at https://www.youtube.com/playlist?list=PLFU7BiIwAjPDqsIL9OLm1z7_RXZA1Jyfj.
>
> We have also found that the distributional model helps most in harder, higher-dimensional tasks. The main characteristic these tasks seem to share is the time/data required with which the solve the task. Potentially due to the complexity of learning the Q-function.
>
> We found in general, for both distributional and non-distributional, that the 5-step version provided better results. Although this is not fully corrected for (see answers to the above reviewers) we found this to experimentally provide quite a bit of benefit to all variations of the algorithm.
>
> We used 51 atoms across all tasks except for the humanoid parkour task which used 101. The level of resolution necessary will depend on the problem under consideration, and controlled by the combination of the number of atoms as well as the V_{min,max} values, however we found this to be relatively robust. Here we changed the number of atoms for the humanoid task in order to keep the resolution roughly consistent with other tasks.
>
> The V_min and V_max values are global values that bound the support of the distribution. However, you are correct that this is what requires the projection. When applying the Bellman operator to a distribution it will more than likely lie outside the bounds given by V_min/V_max, so we project in order to ensure that our distributions are always within these bounds. Again, we also found these values to be relatively robust and we generally set these given knowledge of the maximum immediate reward of the system.
>
> We did not extensively experiment with increasing the exploration noise, but from preliminary experiments we saw that the algorithm was fairly robust to this value. Deviating from the values we used did not significantly hurt nor hinder the algorithm’s performance.
>
> The poor performance of DDPG in these experiments is primarily due to the fact that DDPG is quite slow to learn. For the easier control suite tasks DDPG is actually a feasible algorithm if given enough time. However for the harder tasks (any of the humanoid tasks, manipulator, and parkour tasks) DDPG would take much too long to work effectively. Finally, one of the bigger problems DDPG has is that it can exhibit quite unstable learning which is not exhibited by D4PG.
>
> The easy-ness of the humanoid stand task is more due to the fact that it has less complicated motions to make than any of the other humanoid tasks.
>
> The wall-clock times are for 32 cores on separate machines. We found communication across machines to be fast enough that having them all be on the same machine was not a requirement.
>
> We apologize that the description of the network architecture was poorly explained and will correct it. The networks have two branches, one of which process the the terrain info to produce a lower-dimensional hidden state before combining it with the proprioceptive information. Utilizing this second branch to process the proprioceptive information and reduce it to a smaller number of hidden units is what we refer to as “reducing its dimensionality” however we will explain this better.
>
> We will also explain critic architecture and what we refer to as “heads” further. Here we refer to the “distributional output” component of the network as a head. In this way we can replace the Categorical output with a Mixture of Gaussians output as described in section A. By “head” we only mean this final component which takes the last set of hidden units, passes them through a linear layer, and outputs the parameters of a distribution.

---

### Official Review · AnonReviewer1 · 2017-11-27
**good evaluation, but lacking in originality**

**Rating:** 6
**Confidence:** 5

**Review:**

The paper investigates a number of additions to DDPG algorithm and their effect on performance. The additions investigated are distributional Bellman updates, N-step returns, and prioritized experience replay.

The paper does a good job of analyzing these effects on a wide range of continuous control tasks, from the standard benchmark suite, to hand manipulation, to complex terrain locomotion and I believe these results are valuable to the community.

However, I have a concern about the soundness of using N-step returns in DDPG setting. When a sequence of length N is sampled from the replay buffer and used to calculate N-step return, this sequence is generated according a particular policy. As a result, experience is non-stationary - for the same state-action pair, early iterations of the algorithm will produce structurally different (not just due to stochasticity) N-step returns because the policy to generate those N steps has changed between algorithm iterations. So it seems to me the authors are using off-policy updates where strictly on-policy updates should be used. I would like some clarification from the authors on this point, and if it is indeed the case to bring attention to this point in the final manuscript.

It would also be useful to evaluate the effect of N for values other than 1 and 5, especially given the significance this addition has on performance. I can believe N-step returns are useful, possibly due to effectively enlarging simulation timestep, but it would be good to know at which point it becomes detrimental.

I also believe "Distributional Policy Gradients" is an overly broad title for this submission as this work still relies on off-policy updates and does not tackle the problem of marrying distributional updates with on-policy methods. "Distributional DDPG" or "Distributional Actor-Critic" or variant perhaps could be more fair title choices?

Aside from these concerns, lack of originality of contributions makes it difficult to highly recommend the paper. Nonetheless, I do believe the experimental evaluation if well-conducted and would be of interest to the ICLR community.

---

> ### Author Response · Authors · 2017-12-21
> **Response to AnonReviewer1**
>
> Thanks for the helpful review!
>
> The reason for our use of N-step returns is it allows us to compute the returns as soon as they are collected and insert into replay without storing full sequences. This is done for efficiency reasons. For N>1 this ignores the difference between the behavior and target policies. This could be corrected using an off-policy correction such as Retrace (Safe and Efficient Off-Policy Reinforcement Learning, Munos et al., 2016) but that would require storing full trajectories.
>
> However, for reasonably small N this difference is not great, which is what we show in our experiments. With N much larger than the value of 5, we see a degradation in performance for exactly this reason. We include further discussion of exactly this point.

---

### Official Review · AnonReviewer3 · 2017-11-27
**the proposed method is simple**

**Rating:** 5
**Confidence:** 4

**Review:**


Comment: The paper proposes a simple extension to DDPG that uses a distributional Bellman operator for critic updates, and introduces two simple modifications which are the use of N-step returns and parallelizing evaluations. The method is evaluated on a wide variety of many control and robotic talks.

In general, the paper is well written and organised. However I have some following major concerns regarding the quality of the paper:

- The proposal, D4PG, is quite straightforward which is simply use the idea of distributional value function by Bellemare et al. (previously used in DQN). Two modifications are also simple and well-known techniques. It would be nicer if the description in Section 3 is less straightforward by giving more justifications and analysis why and how distributional updates are necessary in the context of policy search methods like DDPG.

- A positive side of the paper is a large set of evaluations on many different control and robotic tasks. For many tasks, D4PG performs better than the variant that does not use distributional updates (D3PG), however by not much. There are some tasks showing no-difference. On the other hand, the choice of N=5 in comparisons is hard to understand and lacks further experimental justifications. Different setting and new performance metrics (e.g. data efficiency, number of episodes in total) might also reveal more properties of the proposed methods.



* Other minor comments:

- Algorithm 1 consists of two parts but there are connection between them. It might be confused for ones who are not familiar with the actor-critic framework.

- It would be nicer if all expectation operators in Section 3 comes with corresponding distributions.

- page 2, second paragraph: typos in "hence my require less samples to learn"

- it might be better if the reference on arXiv should be changed to relevant publication conferences with archival proceeding: work by Marc G. Bellemare at ICML 2017

---

> ### Author Response · Authors · 2017-12-21
> **Response to AnonReviewer3**
>
> Thank you for the review!
>
> As to the necessity of the distributional updates, the DPG algorithm relies heavily on the accuracy of the value function estimate due to the fact that the gradient computed under the DPG theorem is based only on gradients of the policy pi and gradients of the Q-function. By better estimating the Q-function we directly impact the accuracy of the policy gradient. We will include further discussion of this.
>
> It is true that the distributional version (D4PG) does not always out-perform the non-distributional version (D3PG). However this is typically on easier tasks. In the control suite of tasks the distributional version significantly out-performs on the acrobot, humanoid, and swimmer set of tasks. For manipulation tasks this holds for the hardest pickup and orient task. And finally for all parkour tasks. So for tasks that are already somewhat easy to solve there are limited immediate gains, but for harder tasks this update tends to help (and help significantly for the parkour tasks).
>
> The choice of a higher N is suggested by algorithms such as A3C and Rainbow, among others. Note that the Rainbow algorithm (Rainbow: Combining Improvements in Deep Reinforcement Learning, Hessel et al, 2017) utilizes an off-policy Q-learning update with uncorrected n-step returns, in a very similar way to that used by D4PG. In order to fully correct for this we should be using an off-policy correction, which we have not used for reasons of efficiency (see our response to the next reviewer). However, experimentally we have shown that this minor modification helps quite significantly and can be used directly in any off-policy algorithm. In all of our experiments and across both distributional and non-distributional updates it tends to be better to use the higher N. We did find that increasing N much higher than N>5 tended to degrade performance, which makes sense as this would be more off-policy. We will include further discussion of this aspect of the algorithm.

---

### Public Comment · (anonymous) · 2017-11-05
**Experimental result and set-up**

Hi,

Can you clarify a few questions about experimental set-up and results? In section 4 Result you've describe a few important design choices:

1) You've chosen a fixed Gaussian noise for exploration and made a statement that Ornstein-Uhlenbeck didn't add to performance in your experiments. which contradicts a known results for DDPG. Can you provide a comparison plots and describe an experimental set-up supporting this statement for D4PG?
2) You've chosen the same learning rate for actor and critic, what is a bit different from common practice for DDPG, when a critic usually has leaning rate an order of magnitude higher the same for actor. What is a justification of such a choice? Can provide some experimental results showing performance of a few different choices of learning rates for actor and critic?

In 4.3, Parkour section you showed results of the evaluation of different variants of D4PG and D3PG and comparison vs PPO. But it's unclear what a performance level do they correspond:

3) Can you provide a few videos of the final performance for best variants of D4PG and PPO for Walker2d and Humanoid?
4) You compare with PPO in wall time and number of actors steps. But they are not the only possible metrics for comparison. Can you provide some insights on:
        a) Maximum rewards: What is a maximum performance D4PG vs PPO? How maximum rewards achieved with this 2           algorithm can be compared if perform longer training?
        b) Stability: PPO is known to be very stable algorithm, while DDPG is much more sensitive to the hyperparameters choice. How a stability of D4PG training can be compared to PPO?
        c) Scalability: How well D4PG is scaling with increasing  number of parallel workers in comparison with PPO?
5) Do you plan to release your implementation for D4PG?

---

> ### Public Comment · (anonymous) · 2017-11-22
> **Experimental setup questions**
>
> 1) I had seen some convergence issues when I implemented something similar. Did you face anything similar? How important was the power of the neural approximator and the size of the distribution support set (in case of multinomial distribution)?
>
> 2) Does the extra 'distributed' improve speed or quality of convergence?

---

> > ### Public Comment · (anonymous) · 2017-12-03
> > **New questions**
> >
> > I suppose it should have been questions not to my comment but to the authors of the paper?
> >
> > Also I'd like to refresh my question to authors - do you plan to release a videos showing parkour and robotic hand training results? It's almost a standard for RL research and lack of them can cause some unnecessary suspects and questions.
> >
> > In addition videos can show quality of trained policies.

---

> ### Author Response · Authors · 2017-12-21
> **Response to comment**
>
> We have found that D4PG was very stable and robust to its hyperparameter settings. We generally found that carefully tuning the learning rates was unnecessary, and this also allowed us to eliminate the Ornstein-Uhlenbeck noise.
>
> As to the results on parkour: we have not yet re-run the experiments on the humanoid, however for the 2d-walker these results are approximately at their maximum. And we can see that D4PG outperforms PPO in this setting.
>
> With regards to stability it is well known the DDPG can be quite unstable. As noted above we don’t really see any of these issues with D4PG and it is in fact very stable, both in terms of its behavior during a run, across different seeds, and across different settings of hyperparameters. We haven’t significantly experimented with scaling the number of actors for D4PG, however while we do tend to see performance improvements as we increase the number of workers, we kept this number fixed with the number used in PPO.
>
> Finally, videos of the parkour performance can be found at: https://www.youtube.com/playlist?list=PLFU7BiIwAjPDqsIL9OLm1z7_RXZA1Jyfj.

---

> > ### Public Comment · (anonymous) · 2018-01-10
> > **Parkour performance**
> >
> > Thanks for sharing a video!
> >
> > I have a few questions - a performance looks more poor compare to the original PPO results. Also I've noticed that more challenging environments aren't present. Did you investigate what is a reason of more poor behavior? Can PPO achieve large final reward, but is more slower at start? Or with PPO you've done more experiments and and have larger choice from the best.
> >
> > I would be bery good to have this analysis in the paper - to compare maximum possible reward for PPO and D4PG and in addition distribution of it's final values for training with different seeds. How maximum rewards are compared? Mean? Variance? Not only with the same number of steps like in the paper.

---

### Decision · Program_Chairs · 2018-01-29
**ICLR 2018 Conference Acceptance Decision**

**Decision:**

Accept (Poster)

**Comment:**

As identified by most reviewers, this paper does a very thorough empirical evaluation of a relatively straightforward combination of known techniques for distributed RL. The work also builds on "Distributed prioritized experience replay", which could be noted more prominently in the introduction.